# Exon-18-EGFR Mutated Transformed Small-Cell Lung Cancer: A Case Report and Literature Review

Nunzio Digiacomo [1,*], Tommaso De Pas [1], Giovanna Rossi [1], Paola Bossi [2], Erika Stucchi [3], Fabio Conforti [1], Emilia Cocorocchio [1], Daniele Laszlo [1], Laura Pala [1], Emma Zattarin [1] and Chiara Catania [1]

[1] Department of Medical Oncology, Cliniche Humanitas Gavazzeni, 24125 Bergamo, Italy; giovanna.rossi@gavazzeni.it (G.R.); fabio.conforti@gavazzeni.it (F.C.); emilia.cocorocchio@gavazzeni.it (E.C.); daniele.laszlo@gavazzeni.it (D.L.); laura.pala@gavazzeni.it (L.P.); emma.zattarin@gavazzeni.it (E.Z.); chiara.catania@gavazzeni.it (C.C.)

[2] Department of Pathology, IRCCS Humanitas Research Hospital, 20089 Rozzano, Italy; paola.bossi@humanitas.it

[3] Medical Oncology and Hematology Unit, Humanitas Cancer Center, Humanitas Clinical and Research Center, IRCCS-Via Manzoni 56, 20089 Rozzano, Italy; erika.stucchi@cancercenter.humanitas.it

* Correspondence: nunzio.digiacomo@gavazzeni.it

**Abstract:** Small-cell lung cancer (SCLC) transformation from EGFR mutant adenocarcinoma is a rare entity that is considered to be a new phenotype of SCLC. While transformation from adenocarcinoma (ADC) with EGFR exon 19 deletions and exon 21 L858R point mutations has been described, to our knowledge, no cases of transformation to SCLC from exon-18-mutated ADC have been reported. We reported a clinical case of a patient with exon-18-EGFR-transformed SCLC, and we performed a systematic review of the literature.

**Keywords:** transformed SCLC; EGFR; exon 18

## 1. Introduction

*EGFR*-tyrosine kinase inhibitors (TKIs) represent the standard of care for patients with *EGFR*-mutated non-small-cell lung cancer (NSLC). Resistance to EGFR TKI is becoming a major issue [1]. *T790M* mutation represents the main mechanism of resistance to first and second generation (1st- and 2nd-gen) EGFR TKIs, with the phase III RELAY trial showing a post-progression *T790M* rate of 43% [2]. The mechanisms of resistance to the 3rd-gen EGFR TKI osimertinib are instead more heterogeneous. The development of novel EGFR TKIs has led to increasing resistance: on-target resistance, off-target resistance and transformation. High incidence of small-cell (3%), squamous (6%), and pleomorphic (4%) transformation and frequent acquired gene alterations (e.g., *MET* amplification 6%, *ALK* fusion 6%, *HER2* mutation 3%, *K-RAS* mutation 3%) are also found after treatment with osimertinib [3]. Transformation from NSCLC to small-cell lung cancer (SCLC) represents one of the biological mechanisms of resistance to EGFR TKI, accounting for 3 to 15% of the cases (3 to 5% for cases of progression to 1st- and 2nd-gen EGFR TKI and up to 15% for those progressing on 1st-line osimertinib) [4,5]. At the time of disease progression, tissue biopsy can effectively identify not only on-target or off-target *EGFR*-resistance mutations but also any histological transformation that may occur [6]. Transformed SCLCs are associated with a more aggressive clinical behavior and a poorer prognosis than classic SCLC, and treatments are still controversial and not established yet. Based on retrospective data and case reports in the literature, chemotherapy with platinum and etoposide represents the first-choice treatment, and taxanes also showed a high response rate [7–9]. To our knowledge, no cases of transformation from *EGFR* exon-18-mutated adenocarcinoma (ADC) to SCLC have been reported in the literature. In this study, we report a clinical case of a patient with exon-18-*EGFR*-mutated transformed SCLC, and we performed a systematic analysis of the literature. (See Section 1).

## 2. Clinical Case

In this study, we propose a clinical case that shows how we treated and the effectiveness of treatments of a patient affected by an adenocarcinoma lung positive to exon-18-*EGFR* mutation, who developed SCLC transformation as a resistance mechanism to EGFR-TKi.

To our knowledge, no cases of SCLC transformed by exon-18-mutated ADC have been reported in the literature.

In September 2021, due to a persistent dry cough, a 50-year-old woman with a 30 pack-year smoking history underwent chest x-ray and computed tomography (CT), which showed a lesion of 45 mm in the lower right lung lobe with a 13 mm lymphadenopathy in the ipsilateral mediastinal hilum, 2 suspicious hepatic lesions of 6 and 10 mm, and a focal osteolytic lesion of 5 mm to the right acetabulum. The positron emission tomography (PET) imaging showed increased glucose uptake in all lesions except the liver lesions.

The patient underwent endobronchial ultrasound-guided transbronchial needle aspiration (EBUS-TBNA) on the primary tumor, with diagnosis of *EGFR*-mutated lung adenocarcinoma (Figure 1A). The molecular profile was assessed: an *EGFR* exon 18 [p.L747_S752del; c.2239_2256del18]) mutation was found, while *BRAF*, *ALK*, *ROS1*, and *MET* were wild-type; PDL1 expression was <1%. In December 2021, due to a pathological fracture of the distal left humerus, she underwent osteosynthesis and cementation, with histological diagnosis of bone metastases with focal extension to the periosseous soft tissues of the primary lung adenocarcinoma.

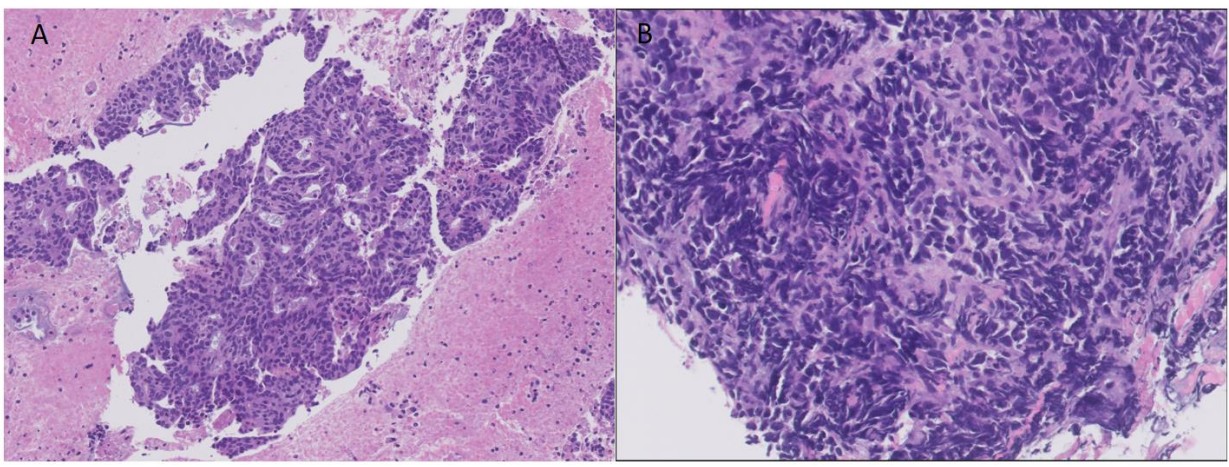

A: Primary Adenocarcinoma before osimertinib administration
Endobronchial Ultrasound – guided transbronchial needle aspiration

B: Transition to SCLC after progression to osimertinib
Sovraclavear node biopsy

**Figure 1.** Tumor biopsy before ((**A**): adenocarcinoma) and after ((**B**): SCLC) progression to osimertinib.

The pathological fracture of the humerus caused a severe functional limitation and intense pain. Radiotherapy was not performed because after an orthopedic consultation, an immediate surgery was suggested. Postoperative radiotherapy was not performed because it was considered not indicated by the radiotherapy team (see Section 1).

Due to the evidence of metastatic disease characterized by the presence of the exon-18-*EGFR* mutation, in December 2021, the patient was started on therapy with osimertinib, and the CT scan in April 2021 showed a good response to treatment, with almost complete regression of both the primary tumor and the right mediastinal hilar lymphadenopathy. At the bone level, a focal osteo-thickening area on the right acetabulum was reported; no osteolytic lesions were found.

The benefit was maintained until August 2022 (approximately 8 months), when the disease progressed (both lung primary lesion and mediastinal hilar lymphadenopathy), and the onset of a right adrenal metastasis and a left-supraclavicular lymph node was documented.

A sovraclavear lymph node biopsy was performed in order to explore any resistance mechanisms to osimertinib, revealing a phenotypic switch to SCLC (Figure 1B): the

morpho-phenotypic findings are referable to localization of neuroendocrine carcinoma with small-cell aspects. Immunocytochemical analyses: CKPool (+); TTF1 (+); Synaptophysin (+); Chromogranin (+); Ki67 (70%)]. NGS analysis was performed, and PDL1 expression was assessed, showing the presence of an identical *EGFR* mutation of exon 18 [p.L747_S752del] to that found at the diagnosis and the presence of a *PIK3CA* mutation [p.E545K, c.1633G > A]; PD-L1 was negative [TPS < 1%].

On 17 August 2022, the patient started chemoimmunotherapy with carboplatin AUC 5–etoposide [100 mg/m$^2$ die 1-2-3] and atezolizumab flat dose 1200 mg, q3 weeks.

A CT scan that was performed on 12 October 2022 at the first radiological assessment after three cycles of treatment (from 17 August 2022 to 28 September 2022) (see Section 2) showed partial response of all known metastatic target lesions, but a new small lesion of 6 mm was found in the left frontal lobe. In consideration of the clinical benefit and of the good overall response of the disease, the patient continued the treatment and a brain MRI examination was performed in view of possible Stereotactic Body Radiation Therapy (SBRT) on the brain metastatic lesion. The brain MRI showed multifocal brain metastases. Patients underwent Whole-Brain Radiation (from 9 January 2023 to 19 January 2023, 27 Gy in 9 fractions, 3D-CRT technique with X photons 6–18 MV from Varian Trilogy) (see Sections 1 and 2).

On 6 February 2023, six months after diagnosis, a radiological assessment of disease by CT scan was carried out, which showed significant progression of the disease in the brain, lung, and abdomen. A re-biopsy of the metastatic lesions was scheduled; however, it could not be performed due to the deterioration of the patient's clinical condition. In consideration of the worsening of the patient's performance status, best supportive care was activated.

## 3. Discussion and Conclusions

Transformation from *EGFR*-mutant NSCLC to SCLC during treatment with EGFR-TKI represents an important mechanism of acquired resistance.

While histological transformation from adenocarcinoma (ADC) with *EGFR* exon-19 deletion and exon-21 L858R point mutation has been described, to our knowledge, no cases of exon-18-mutated ADC have been reported. Moreover, in the literature, there are no studies focused on the efficacy of chemoimmunotherapy combination for patients affected by transformed SCLC. Very limited data from small retrospective studies which assess the activity of immunotherapy as monotherapy are available.

The retrospective study by Marcoux et al. on 67 patients—58 with NSCLC and 9 with de novo SCLC—in which only NSCLC patients and none of the SCLCs received EGFR-TKI, showed that median time to transformation was 17.8 months. After transformation, platinum-etoposide and taxane regimens were the most frequently prescribed and associated with better objective responses of 54 and 50%, respectively. Among taxanes, paclitaxel and nab-paclitaxel were associated with a high response rate of up to 71%, while docetaxel was associated with a 0% response rate. Median PFS was 3.4 months and 2.7 months with platinum-etoposide and taxanes, respectively. Following transformation, the median survival for SCLC patients was 10.9 months. Interestingly, among 17 patients who had received immunotherapy, including anti-PD1, anti-CTLA-4 agents, or combinations thereof, none of them had a clinical response, suggesting that these tumors are more similar to *EGFR* mutant NSCLC, which are known to be characterized by a poor response to immunotherapy compared to classic SCLC cases. [8]. A systematic review by Roca et al. on 39 patients with diagnosis of lung adenocarcinoma, of which 16 had received 1st- and 2nd-generation TKIs and 1 had received osimertinib, showed a median time to transformation of 19 months. After SCLC transformation, patients were treated with platinum-etoposide chemotherapy or combined chemoradiotherapy, and a median survival of 6 months for patients who had received 1st- and 2nd-generation TKIs and 2 months for those who were treated with osimertinib were observed. The multivariate analysis demonstrated that smoking status was significantly associated with poorer survival, and female sex was associated with longer time to SCLC transformation [10].

In clinical practice, the efficacy of immunotherapy for patients with transformed SCLC remains unclear. Reports by Nishikawa et al. and Tokaca et al. showed that there is no clinical benefit with immunotherapy with nivolumab, as did the retrospective study by Marcoux, in which patients received treatments with anti-PD1, anti-CTLA4, and anti-PDL1 agents with no clinical improvement [8,11,12].

Some reports have suggested an important relationship between oncogenic signaling linked to *EGFR* mutation and PDL1 expression, demonstrating that treatment with EGFR TKIs may cause the downregulation of PD-L1 expression [11,13,14]. These studies demonstrated that inhibition of a PD-1 pathway by anti-PD1 antibodies is associated with lower levels of tumor-promoting cytokines and that *EGFR/ALK-* inhibitors are associated with significant downregulation of PDL1 expression [13,14].

After disease progression, due to transformation to SCLC, the benefit of continuing therapy with osimertinib or other *EGFR-* TKI inhibitors in combination with chemotherapy remains unclear.

A case report published by Leonetti et al. showed that osimertinib in addition to platinum-etoposide chemotherapy was effective in inhibiting cell growth on cell cultures from post-osimertinib biopsy in a patient affected by NSCLC with T790M-EGFR mutation who received second-line osimertinib and developed disease progression linked to SCLC transformation. This finding suggests that TKI inhibitors are an effective therapeutic option [9].

Another interesting example confirming the utility of maintaining TKI inhibition in association with chemotherapy can be found in Batra's publication, which reported a case of a patient affected by *EGFR*-mutant SCLC and lung ADC who obtained a good response to therapy with osimertinib [15].

Although SCLC transformation has been reported mostly in *EGFR*-mutated adenocarcinomas as a resistance mechanism to EGFR TKI, it can occur less frequently in patients with *EGFR* wild-type ADC who were not exposed to TKIs.

A study by Ferrer et al. including 48 *EGFR*-mutated and 13 non-mutated NSCLC patients showed that the median time to transformation was 16 months in the *EGFR*-mutant group, compared to 26 months in the non-mutant ones. This could be linked to the increased aggressiveness of the mutated disease [16]. Moreover, from a biological point of view, we can hypothesize that the strong inhibition of the activated *EGFR* pathway by *EGFR-* TKIs may cause tumor cells to trigger rapid escape mechanisms and, consequently, cause early clinical progression. For example, this mechanism has been widely described in the literature to explain the ineffectiveness of BRAF inhibitors in metastatic colorectal cancer (mCRC) cells, which is the opposite of what happens in melanoma. In fact, in mCRC, based on the well-known biological phenomenon of "paradoxical activation of the MAPK pathways", an abnormal re-activation of the MAPK pathway causes up-stimulation of cell proliferation, which is triggered by the action of a single-agent BRAF inhibitor [17].

Compared to the *EGFR* wild-type group, patients with *EGFR*-mutated tumors showed a partial response rate of 45% to platinum-etoposide chemotherapy (versus 40%); a shorter median overall survival since initial diagnosis of 28 months (versus 37 months), as expected due to the higher clinical aggressiveness of *EGFR-* and *ALK*-mutated tumors; and a similar median overall survival since time of transformation: 9 months (versus 10 months) [16].

Other retrospectives studies by Ahmed et al. and Ahn et al. showed that SCLC transformation can occur in both *EGFR*-mutant and *EGFR*-wild-type NSCLCs, as well as both adenocarcinoma and squamocellular carcinoma, suggesting that this may be a common mechanism of acquired resistance to several kinds of treatments, not only EGFR TKI [18,19]. (See Section 1).

In line with the literature, our patient received platinum-etoposide chemotherapy as the first-choice treatment for transformed SCLC. Based on the demonstrated efficacy of a chemo-immunotherapy combination as a standard first-line treatment for classic SCLC, we added atezolizumab.

From 17 August 2022 to 28 September 2022, the patient underwent three cycles of treatments. On 12 October 22, at the first radiological assessment via CT after three cycles, we observed a significant overall response of the disease, indicating that it was sensitive to treatment. Several published studies demonstrated a significant association between heavy smoking status and elevated tumor PD-L1 expression, with better results of anti-PD-1 therapy in smoker patients with advanced NSCLC compared to non-smokers. Tobacco smoking is associated with T-cell exhaustion, the upregulation of PD-1, and high Tumor Mutational Burden (TMB) in tissue and blood. Therapeutic inhibition of PD-1 and PD-L1 interactions can lead to restored T-cell function and anti-tumor activity, and TMB has been raised as a potential predictive marker of benefit for anti-PD1 agents. (See Section 3) [20–22]. Interestingly, our patient has a history of heavy exposure to cigarette smoke. Although the analysis of PDL1 expression at the initial diagnosis of NSCLC was negative and was not repeated at the time of disease progression and transformation, it cannot be excluded that several molecular mechanisms strongly associated with smoking may contribute to the significant response obtained to the chemo-immunotherapy treatment, which, in this case, contains an anti-PDL1 agent.

Based on this emerging evidence of an association between immune system efficacy and smoking history, we hypothesize that the combination of chemotherapy and immunotherapy may be effective for patients with a history of heavy smoking. For our patient, time to progression from initial NSCLC diagnosis during osimertinib therapy was approximately 8 months, shorter than what is reported in the literature. OS since transformation in SCLC cannot be define yet because treatment is ongoing.

Although EGFR TKIs represent the standard first-line therapy for EGFR-mutated NSCLC, further efforts are needed to elucidate the relationship between the resistance mechanisms related to *EGFR* mutations that occur during therapy with EGFR-TKI, and the complex immunotherapy resistance in order to improve the efficacy of treatments for lung cancer, including very rare entities such as transformed SCLC, which represents a very aggressive tumor in the context of *EGFR*-mutated NSCLC.

De novo small-cell lung cancer and transformed SCLC from *EGFR*-mutant NSCLC are two different entities with regard to theirbiological behavior, response to treatments, and prognosisFurthermore, additional complexity is due to the existence of different *EGFR* mutations. In fact, very little is known about the so-called uncommon variants that may lead to important differences in both downstream signaling networks and resistance mechanisms, such as histological transformation, compared to the common *EGFR* mutations.

**Author Contributions:** Conceptualization, C.C., N.D. and T.D.P.; methodology, N.D., T.D.P., G.R., P.B., E.S., F.C., E.C., D.L., L.P., E.Z. and C.C.; Investigation, C.C. and N.D.; data curation: P.B., G.R., F.C., L.P., E.Z. and E.S.; writing—original draft preparation, N.D. and C.C.; writing—review and editing: N.D., T.D.P., G.R., P.B., E.S., F.C., E.C., D.L., L.P., E.Z. and C.C.; visualization and; supervision, N.D., C.C. and G.R. All authors have read and agreed to the published version of the manuscript.

**Funding:** This research received no external funding.

**Institutional Review Board Statement:** Not applicable.

**Informed Consent Statement:** Informed consent was obtained from the subject involved in the study.

**Data Availability Statement:** Not applicable.

**Conflicts of Interest:** The authors declare no conflict of interest.

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
