# Peer review of "Exon-18-EGFR Mutated Transformed Small-Cell Lung Cancer: A Case Report and Literature Review"

_curroncol, doi:10.3390/curroncol30030265_

Round 1

Reviewer 1 Report

The authors report a clinical case of a patient with exon 18 EGFR-transformed SCLC. The authors suggest to have performed a "systematic review" of literature, but no of the PRISMA criteria for a systematic review  were met (no inforrmation about search terms, databases used, number of identified articles, inclusion and exclusion criteria, etc.) The manuscript contains several typographical (or linguistic) errors (e.g. “sovraclavear” instead of “supraclavicular”, “surrenalic” instead of “suprarenal”, “April 2021” instead of apparetly April 2022, or “from 08/17/2022 to 09/28/2022” etc. The reported cases should be summarized in a table. The exact dates may allow the patient's identity to be undetected. Has consent been obtained from the patients or their family? If not, the precise dates should be replaced with time intervals to hide the patient's identity. The last documented follow up took placein September 2022. SCLC is a rapidly progressive disease - and the current time is February 2023. Please update the report. The case description would benefit from Figures which would illustrate the disease course and histological findings. The reference format does not meet the mdpi requirements.  I do not feel that the paper would meet the criteria for publication in its current form.

Author Response

Dear Reviewer,

We wanted to thank you for your comments and suggestions, that definitely contributed to improve our paper.

We modified our manuscript accordigly.

Moreover, we performed a revision for language and grammar.

Thank you very much

Reviewer 2 Report

In this paper Digiacomo and colleagues report an interesting case of a LUAD converting to SCLC in the course of therapy. Before publication some major issues must be addressed.

General:

·         Add more information on additional treatment such as radiation (e.g. figures on radiation ports) and images of histological sections.

·         The paper has to be reviewed by an English speaking native. Some parts are unclear to the reader because of major language flaws.

Title:

·         As there are no reports specifically dealing with the transformation of a EGFR-mutated LUAD to SCLC leave out and Literature Review

Introduction:

·         Please shorten this considerably and transfer the literature evaluation to the discussion section of the paper (e.g.: the paragraph on the case report published by Leonetti)

·         Please discuss previous studies in the context of the current case.

Clinical case:

·         The authors say that the patient underwent surgery for bone fracture of the left upper arm. Did she not receive post-operative radiation therapy as would be expected? Please clarify.

·         Last paragraph on page 3: Did she receive SABR or will she undergo SABR for the cerebral lesion?

Discussion:

·         The discussion is largely a repetition of the case report. Some of the literature mentioned in the introduction should be discussed here in the context of the case (see above).

·         5th paragraph on page 4 At first radiological assess … Could the authors please establish a closer context with the proposed case?

·         6th paragraph on page 4: What are the molecular mechanisms that predispose the microenvironment to a more efficient answer to immunotherapy in heavy smokers as opposed to non-smokers or ex-smokers? Please explain and quote literature.

·         Last but one paragraph: Calling this transformed SCLC a new entity seems very far-fetched since it is highly unlikely that in primary diagnosis the tumor was of mixed histology and the SCLC part was just missed out.

Author Response

Dear Reviewer,

We wanted to thank you for your comments and suggestions, that definitely contributed to improve our paper.

We modified our manuscript accordigly .

General:

1.Add more information on additional treatment such as radiation (e.g. figures on radiation ports)

Thank you for your suggestion. We added the following  information:

DONE - The pathological fracture of the humerus conditioned a severe functional limitation of the limb and intense pain symptoms. Radiotherapy was not performed after orthopedic consultation which indicated im-mediate surgery. Postoperative radiotherapy was not performed because it was not specif-ically requested by the radiotherapy team.

DONE - Patients was underwent to Whole Brain Radiation (From 09/01/2023 to 19/01/2023 dose 27 Gy in 9 sittings, 3D-CRT technique with X photons 6-18MV from Varian Trilogy).

          1a   images of histological sections -  DONE

  1. The paper has to be reviewed by an English speaking native. Some parts are unclear to the reader because of major language flaws.

DONE - 

Thank you for your suggestion. The paper was revised accordingly.

Title:

  1. As there are no reports specifically dealing with the transformation of a EGFR-mutated LUAD to SCLC leave out and Literature Review:

Thank you for your comment. While we understand your point, we feel that we have now added more references to the literature, as reported below and following your suggestions (see below:  Introduction, question 2. ).

Introduction:

  • Please shorten this considerably and transfer the literature evaluation to the discussion section of the paper (e.g.: the paragraph on the case report published by Leonetti)

DONE –  We moved the literature evaluation to the discussion section as you suggested

The retrospective trial by Marcoux at al. on 67 patients, 58 NSCLC and 9 SCLC de novo, in which only NSCLC patients and none of the SCLCs were underwent to EGFR-TKI, showed that median time to transformation was 17.8 months. After transformation, platinum-etoposid and taxanes regimens were the most used treatments and associated with better objective responses, 54 and 50% respectively. Among taxanes, paclitaxel and nab-paclitaxel were associated with high response rate up to 71%, while docetaxel was associated with 0% response rate. Median PFS was 3.4 months and 2.7 months with platinum-etoposide and taxanes, respectively. Since trasformation, the median survival for SCLC patients was 10.9 months. Interestingly, among 17 patients who had received immunotherapy, including anti-PD1, anti-CTLA-4 agents, or combinations) none of them had a clinical response, suggesting that these tumors are more similar to EGFR mutant NSCLC, which are known to be characterized by a poor response to immunotherapy, compared to classic SCLC cases. [8]. A systematic review by Roca at al. of 39 patients with lung adenocarcinoma at diagnosis, of which 16 had received 1rd and 2rd generation TKIs and one had received osimertinib, showed a median time to transformation of 19 months. After SCLC transformation, they were treated with platinum-etoposid chemotherapy or combined chemoradiotherapy showing a median survival of 6 months for patients had received 1st and 2nd generation TKIs and 2 months for those had treated with osimertinib. The multivariable analysis found that smoking status was significantly associated with poorer survival and female gender was associated with longer time to SCLC transformation [10].

In clinical setting, the efficacy of immunotherapy remains unclear to treat patients with transformed SCLC. The reports by Nishikawa at al. and Tokaca at al., showed that immunotherapy with nivolumab had no clinical response, as well the retrospective analyses of Marcoux where patients underwent to treatments with anti-PD1, anti CTLA4 and anti PDL1 had no clinical benefit [11, 12, 8].

Some reports have suggested an important relationship between oncogenic signaling linked to EGFR mutation and PDL1 expression, demonstrating that the therapy with EGFR TKIs may cause the downregulation of PD-L1 expression. [11, 13, 14]. These studies have demonstrated that inhibition of PD-1 pathway by anti-PD1 antibodies is associated with lower levels of tumor-promoting cytokines and that EGFR/ALK- inhibitors are associated with significant downregulation of PDL1 expression [13 – 14].

After disease progression due to transformation of SCLC, continuing therapy with osimertinib, or other EGFR- TKI inhibitors, in combination with chemotherapy remains unclear.

A case report published by Leonetti at al. showed that osimertinib in addition to platinum-etoposid chemotherapy was effective to inhibit cells growth on cell cultures from post-osimertinib biopsy in a patient affected by NSCLC with T790M-EGFR mutation underwent to second-line osimertinib who developed progression disease linked to SCLC trasformation, suggesting how the maintenance of molecular target inhibition with TKI inhibitor drugs could be an effective therapeutic option [9].

Another interesting example confirming the possible usefulness of maintaining the receptor block with TKI inhibitors associated with chemotherapy treatment this can be seen from Batra's publication which reported a case of patient affected by EGFR-mutant SCLC and lung adenocarcinoma who showed good response to therapy with osimertinib [15].

Although SCLC trasformation has been reported mostly in adenocarcinomas EGFR mutant as resistance mechanism to EGFR TKI therapies, less frequently it can occur in patients without EGFR mutations do not expose to TKIs and with different histologic characteristics.

The study of Ferrer at al. including 48 EGFR mutant and 13 non- mutant NSCLC patients showed that the median time to transformation was 16 months in the EGFR-mutant group, compared to 26 months in the non-mutant ones. This could be linked to the increased aggressiveness of the mutated group [16].  Moreover, from a biological point of view, we can hypothesize that the strong inhibition of the activated EGFR pathway by EGFR- TKI may cause tumor cells to trigger rapid escape mechanisms and consequently cause early clinical progression. For example, this mechanism has been widely described in the literature to explain the ineffectiveness of BRAF inhibitors in mCRC cells unlike what happens in melanoma, based on the known biological phenomenon of "paradoxical activation of the MAPK pathways" as which abnormal re-activation of MAPK pathway through several kinases causing up-stimulation of proliferation cell signals, triggered by the action of BRAF inhibitor as single agents [17].

Compared to EGFR non- mutant group, the EGFR mutant patients showed a partial response rate of 45% to platinum-etoposide chemotherapy (versus 40%), a shorter median overall survival since initial diagnosis of 28 months (versus 37 months), as expected because of the higher clinical and metastatic aggressiveness of EGFR or ALK mutated tumours, and a similar median overall survival since time of transformation, 9 months (versus 10 months) [16].

Other retrospectives studies by Ahmed et al. and Ahn at al. showed that SCLC transformation can occur both EGFR-mutant and EGFR–wild-type NSCLCs, both adenocarcinoma and squamocellular carcinoma, suggesting that it may be an acquired resistance mechanism to several kinds of treatments, not only EGFR TKI treatment and it may be more common than thought [18, 19].

  1. Please discuss previous studies in the context of the current case:

Following your suggestion, we reported and discussed previos studies in the discussion section and in the introduction section.

Clinical case:

  1. The authors say that the patient underwent surgery for bone fracture of the left upper arm. Did she not receive post-operative radiation therapy as would be expected? Please clarify.

We clarified this point in the manuscript, as follow:

DONE - The pathological fracture of the humerus conditioned a severe functional limitation of the limb and intense pain symptoms. Radiotherapy was not performed after orthopedic consultation which indicated immediate surgery. Postoperative radiotherapy was not performed because it was not specifically requested by the radiotherapy team

  1. Last paragraph on page 3: Did she receive SABR or will she undergo SABR for the cerebral lesion?

DONE – We clarified this aspect in the manuscript

Delete: In consideration of the clinical benefit, of the good overall response of disease, the patient continues the treatment and will undergo to brain MRI examination in view of possible Stereotactic Body Radiation Therapy (SBRT) on brain metastatic lesion.

We added: Brain MRI was perfomed that showed multifocal brain metastases. Patients was underwent to Whole Brain Radiation (From 09/01/2023 to 19/01/2023 dose 27 Gy in 9 sittings, 3D-CRT technique with X photons 6-18MV from Varian Trilogy).

Discussion:

  1. The discussion is largely a repetition of the case report. Some of the literature mentioned in the introduction should be discussed here in the context of the case (see above).

Following your suggestion, we moved literature evaluation to the discussion session.

DONE – see your suggest à point 1 of introduction: we transfer literature evaluation to the discussion section.

  1. 5th paragraph on page 4 At first radiological assess … Could the authors please establish a closer context with the proposed case

Thank you for your comment. We added more information.

DONE CT scan performed on 10.12.2022 after three courses of treatment (from 08.17.2022 to 09.28.2022)

  1. 6th paragraph on page 4: What are the molecular mechanisms that predispose the microenvironment to a more efficient answer to immunotherapy in heavy smokers as opposed to non-smokers or ex-smokers? Please  explain and quote literature.

Thank you for your comment. We clarified this aspect and we added references to the literature.

DONE - Several published studies have shown a significant association between heavy smoking status and elevated tumor PD-L1 expression, showing better results of anti-PD-1 therapy in smoking patients with advanced NSCLC compared to non-smokers. Tobacco smoking is associated with T-cell exhaustion, upregulation of PD-1 and high Tumor Mutational Burden (TMB) in tissue and blood. Therapeutic inhibition of PD-1 and PD-L1 interactions can lead to restored T-cell function and anti-tumour activity, and TMB has been raised as potential predictive marker of benefit for anti-PD1 agents.

  1. Last but one paragraph: Calling this transformed SCLC a new entity seems very far-fetched since it is highly unlikely that in primary diagnosis the tumor was of mixed histology and the SCLC part was just missed  out:

Thanks for this suggestion. The excellent response to osimertinib treatment in a patient with a rare EGFR mutation suggests that the patient had pure NSCLC at diagnosis and subsequently developed a transformed SCLC.

Round 2

Reviewer 1 Report

No doubt the authors have made efforts to improve the manuscript. The work became more readable and contains more information. I leave it to the editor's discretion whether this much improved manuscript will be accepted.

Reviewer 2 Report

I have nothing to add.